# Effect of Cold-Rolling Directions on Recrystallization Texture Evolution of Pure Iron

**DOI:** 10.3390/ma15093083

**Published:** 2022-04-24

**Authors:** Toshio Ogawa, Yutaro Suzuki, Yoshitaka Adachi, Atsushi Yamaguchi, Yukihiro Matsubara

**Affiliations:** 1Department of Materials Design Innovation Engineering, Graduate School of Engineering, Nagoya University, Furo-cho, Chikusa-ku, Nagoya 464-8603, Aichi, Japan; suzuki.yutaro.s0@s.mail.nagoya-u.ac.jp (Y.S.); adachi.yoshitaka@material.nagoya-u.ac.jp (Y.A.); 2Asahi-Seiki Manufacturing Co., Ltd., 5050-1 Shindenbora, Asahimae-cho, Owariasahi 488-8655, Aichi, Japan; atsushi-yamaguchi@asahiseiki-mfg.co.jp (A.Y.); yukihiro-matsubara@asahiseiki-mfg.co.jp (Y.M.)

**Keywords:** texture, recrystallization, cold-rolling direction, pure iron

## Abstract

The influence of cold-rolling directions on the recrystallization texture evolution of pure iron was examined. As-received pure iron sheets were cold-rolled under two different conditions (specimens A and B). Specimen A was cold-rolled in the vertical direction against the cold-rolling direction of the as-received sheet. Specimen B was cold-rolled in the vertical direction against the cold-rolling direction of the as-received sheet, and then in the cold-rolling direction of the as-received sheet. Cold-rolled specimens were heated to each desired temperature before being quenched in water to room temperature (298 ± 2 K). Both cold-rolled specimens showed the development of γ-fiber and {100}<011> orientation. Additionally, γ-fiber formed comparatively more in cold-rolled specimen A, while α-fiber developed comparatively more in cold-rolled specimen B. Strain distribution in cold-rolled specimen A was presumably inhomogeneous, whereas that in cold-rolled specimen B was rather uniform at the macro-scale. The formation of γ-fiber was confirmed in annealed specimen A. In annealed specimen B, however, the recrystallization texture tended to be random, and the formation of α-fiber was observed. Furthermore, the formation of Goss orientation in both annealed specimens was established. Recrystallized ferrite grains with Goss orientation nucleated in high strain regions of cold-rolled specimen. These findings show that by devising the cold-rolling direction, it is possible to discover new types of recrystallization textures.

## 1. Introduction

One way of improving material properties such as mechanical and magnetic properties is to control the texture of metals [1,2,3]. The texture evolution of iron and steel was studied to acquire a high *r*-value as an indication of deep drawability [4,5,6]. The *r*-value is expressed as a ratio of true strain in width to true strain in thickness, and a high *r*-value indicates excellent deep drawability.

The development of {111}//normal direction (ND) fiber texture (γ-fiber) and <110>//rolling direction (RD) fiber texture (α-fiber) in conventional one-way cold-rolled iron and steel is generally confirmed [7]. Following annealing, γ-fiber with high strain develops preferentially while consuming α-fiber with low strain, and eventually only γ-fiber develops. The development of γ-fiber is well known to improve the *r*-value. The grain refinement in the initial microstructure [4,5], the reduction in the amount of solute carbon before cold-rolling [5,6], and the rise in the cold reduction rate before annealing [5,6] are all crucial in the development of the γ-fiber. Yoshinaga et al. [8] created electro-deposited pure iron with an extremely high *r*-value. Additionally, the texture formation behavior is affected by the addition of alloying elements and annealing conditions. For instance, we demonstrated that when Nb-added low-carbon steel is heated rapidly, recrystallized ferrite grains with α-fiber connecting along with the RD are observed [9]. On the other hand, it has been shown that the development of {110}<001> (Goss orientation) increases *r*-value anisotropy [10]. Recrystallized grains with the Goss orientation prefer to nucleate from shear bands formed by cold-rolling [11]. Thus, the development of Goss orientation is rare in pure iron, whereas it is common in carbon steels because of the presence of carbon, which causes the formation of shear bands as a result of cold-rolling [11]. Previous research suggests that the absence of carbon (e.g., pure iron and interstitial free (IF) steel) results in the formation of γ-fiber and the inhibition of Goss orientation, resulting in a high *r*-value.

It has recently been reported that even in pure iron, the production of a characteristic texture other than the γ-fiber has occurred [7,12]. For example, Tomita et al. [7] revealed that when pure iron is cold-rolled at a reduction rate of 99.8%, the extreme development of α-fiber is observed. Furthermore, Okai et al. [12] found that two-stage cold-rolling and annealing produces a near-cube orientation. Previous studies imply that cold-rolling conditions can produce a variety of textures, even in pure iron. In order to attempt further improvement of material properties in pure iron, it is crucial to investigate the possibilities of producing diverse textures. Gobernado et al. [13] demonstrated in an example of IF steel that cross-rolling produces {100}<011> orientation and subsequent annealing produces {311}<136> orientation. However, the effect of cross-rolling on recrystallization texture evolution during annealing in pure iron has not been studied. Furthermore, in an earlier study [13], the mechanism of recrystallization texture evolution was not fully understood. As a result, the goal of this research is to discover the possibility of producing diverse textures in pure iron by devising cold-rolling conditions. We focused on the effect of cold-rolling directions on recrystallization texture evolution.

## 2. Materials and Methods

The current investigation made use of as-received pure iron sheets (thickness: 1.4 mm). Figure 1 depicts an optical micrograph and an orientation distribution function (ODF) map of the as-received sheets. The average grain size was 22.0 ± 9.8 μm (Figure 1a), and α-fiber and γ-fiber development were both confirmed (Figure 1b). As shown in Figure 2, the sheets were cold-rolled in two distinct conditions (specimens A and B). Specimen A was cold-rolled to a thickness of 0.56 mm in the vertical direction against the cold-rolling direction of the as-received sheet (a 60% reduction). Specimen B was cold-rolled to a thickness of 0.98 mm in the vertical direction against the cold-rolling direction of the as-received sheet, and then to a thickness of 0.56 mm in the as-received sheet’s cold-rolling direction (a total reduction of 60%). Following cold-rolling, the specimens were heated to each target temperature at a rate of 5 K/s, and then water-quenched to room temperature within 2 s of being removed from the furnace. Some specimens were kept at 823 K for 5 s to study the recrystallization texture evolution.

Microstructure and texture analysis in the RD-ND plane were performed on cold-rolled and annealed specimens utilizing an electron backscatter diffraction/field emission scanning electron microscopy (EBSD/FEGSEM) system and the orientation imaging microscopy (OIM) analysis software. The step sizes for the EBSD measurements ranged from 0.5 to 2.0 μm.

Vickers hardness tests were performed for 10 s with an applied load of 9.8 N to evaluate the recovery and recrystallization process. The standard deviation was calculated based on the results of three specimens.

## 3. Results and Discussion

### 3.1. Analysis of Cold-Rolled Sheets

#### 3.1.1. Microstructural Features of Cold-Rolled Sheets

Figure 3 depicts the image quality (IQ), kernel average misorientation (KAM), and inverse pole figure (IPF) maps of cold-rolled specimens. Grains elongated in the RD and shear bands were found in both specimens, as shown in Figure 3a,e. Although shear band formation has been observed in iron and steel, including alloying elements [14], it has rarely been confirmed in pure iron [15]. Furthermore, in non-oriented electrical steel, cross cold-rolling has been shown to result in a higher dislocation density than conventional cold-rolling [14]. As a result, cross cold-rolling is likely to have strained both specimens more than conventional cold-rolling, resulting in the formation of shear bands.

It should also be noted that the distributions of the KAM maps of both specimens are different (Figure 3b,f). KAM values have been reported to rise as dislocation density (i.e., strain) increases [16,17]. According to previous research, the strain distribution in specimen A is presumably inhomogeneous because some grains have red (high strain) and blue (low strain) colors. In contrast, despite being inhomogeneous at the micro-scale, the strain distribution in specimen B is rather uniform at the macro-scale. Furthermore, the primary texture component was identical in both specimens (Figure 3c,d,g,h). As illustrated in Figure 3b–d, the low KAM value region in specimen A corresponded to the α-fiber. Hashimoto et al. [18] discovered that the dislocation density in deformed grains with α-fiber is insufficient to recrystallize early in IF steels. The findings of this report indicate that the strain in deformed grains with α-fiber is minimal, which is consistent with the results of this investigation. However, no obvious correlation between KAM value and texture was found in specimen B (Figure 3f–h). Furthermore, in cold-rolled specimens A and B, the average KAM value was 1.58 and 1.86, respectively. As previously stated, cross cold-rolling has been linked to increased dislocation density [14]. The results indicate that strain is high in deformed grains with α-fiber, and specimen B is more strained overall than specimen A.

#### 3.1.2. Features of Rolling Texture

Figure 4 depicts two typical ODF maps for each cold-rolled specimen. The development of γ-fiber and {100}<011> orientation was observed in both cases (Figure 4a,c). Gobernado et al. [13] discovered the development of γ-fiber and α-fiber (mainly {100}<011> orientation) in cross-rolled IF steel. This previous study’s finding is consistent with the findings of this investigation. Furthermore, γ-fiber formed comparatively more in specimen A (Figure 4b), while α-fiber developed comparatively more in specimen B (Figure 4d). According to Tomita et al. [7], the extreme development of α-fiber is found when pure iron is cold-rolled at a reduction rate of 99.8%. The α-fiber development has also been reported in severely rolled IF steel [19]. Previous studies suggest that heavy deformation is required for the development of α-fiber. As previously stated, specimen B appears to be more strained overall than specimen A, resulting in the formation of α-fiber in specimen B.

### 3.2. Texture Analysis of Annealed Sheets

Figure 5 depicts optical micrographs of specimens heated to 1073 K. Ferrite recrystallization was completed in both specimens when the temperature reached 1073 K, because no non-recrystallized ferrite grains were observed. Furthermore, the grain size and morphology of the ferrite were very similar in both specimens. Figure 6 depicts the ODF maps of specimens heated to 1073 K. γ-fiber formation was detected in both specimen A and the cold-rolled specimen (Figure 6a). This result is consistent with prior reports [4,5,6]. Furthermore, the formation of Goss orientation was confirmed, despite the fact that it was not visible in the cold-rolled specimen. When compared to the cold-rolled specimen, the recrystallization texture of specimen B tended to be random (Figure 6b). Furthermore, the formation of α-fiber and Goss orientation was established. This section focuses on the formation of Goss orientation. Abe et al. [20] demonstrated that Goss orientation is one of the primary components when the initial grain size in pure iron is high (>80 μm). However, as illustrated in Figure 1a, the grain size prior to cold-rolling was small (approximately 22 μm). Recrystallized ferrite grains with the Goss orientation are also preferentially nucleated in shear bands [21,22,23]. Shear bands are well known to be uncommon in pure iron [7,12,15]. On the other hand, cross cold-rolling leads to higher strain, thereby producing shear bands in this study. As a result, the formation of shear bands is likely to be one of the crucial points in elucidating the mechanism of nucleation of recrystallized ferrite grains with Goss orientation.

Figure 7 depicts the IQ, KAM, and IPF maps for the same area in specimens A, cold-rolled and annealed for 5 s at 823 K. This specimen was in the process of ferrite recrystallization, and the area within the red circle in each map corresponds to Goss orientation. Goss orientation nucleation sites have the following characteristics:(1)Recrystallized ferrite grains with Goss orientation, primarily nucleated in regions with high KAM value.(2)In the cold-rolled specimen, the nucleation sites of recrystallized ferrite grains with Goss orientation coincided with the region of non-recrystallized ferrite grains with γ-fiber and the interface between γ-fiber and α-fiber.

These results imply that recrystallized ferrite grains with Goss orientation nucleated neighboring γ-fiber with higher strain regions via discontinuous recrystallization. Shear bands have been observed to form in deformed grains with γ-fiber [24]. As illustrated in Figure 3a, the shear bands in specimen A were formed within grains with γ-fiber. As a result of the formation of shear bands, recrystallized ferrite grains with the Goss orientation can nucleate. However, these findings can only imply that recrystallized ferrite grains with Goss orientation nucleate in high strain regions. The effect of shear bands on the nucleation of recrystallized ferrite grains with the Goss orientation should be investigated further in the future.

Figure 8 depicts the changes in Vickers hardness as the temperature is raised to 1073 K. Recrystallization began at 823 K and ended at 873 K for both specimens. The temperature at which recrystallization began is consistent with previous studies [15]. Additionally, as previously reported in our study [25], changes in Vickers hardness during heating to 823 K are attributed to recovery progress. During heating to 823 K, the hardness of specimen A dropped slightly, whereas it gradually declined in specimen B. Furthermore, as-cold-rolled specimens A and B had nearly identical hardness. These findings show that specimen B’s recovery progress is faster than specimen A. Cold-rolled specimen B showed the formation of α-fiber with high strain, as shown in Figure 3 and Figure 4. α-fiber has been identified as a low strain deformation texture, and continuous recrystallization from α-fiber has been observed to occur only infrequently [18]. Tomita et al. [7], on the other hand, demonstrated that when cold-rolled pure iron with formed α-fiber is annealed, the texture rarely changes. They propose that continuous recrystallization from severely strained α-fiber is possible. In this study, the recrystallization texture tended to be random and the formation of α-fiber was confirmed in specimen B. Figure 9 depicts the IQ and IPF maps of specimen B after heating to 823 K. Subgrains formed within non-recrystallized ferrite grains with α-fiber were observed. These findings suggest that continuous recrystallization from severely strained α-fiber occurred partially in specimen B, and that the gradual decrease in hardness during the recovery process is due to the acceleration of the continuous recrystallization. Furthermore, it appears that recrystallized ferrite grains nucleated throughout the entire specimen B as a result of the comparably homogeneous strain distribution, resulting in the randomization of the recrystallization texture.

The formation of γ-fiber and/or α-fiber as recrystallization textures in pure iron has been described. However, by devising the cold-rolling directions, we discovered the manufacturing conditions for developing the various textures (especially α-fiber and Goss orientation) in pure iron. Furthermore, initial microstructures such as grain size, texture, and precipitate, and/or inclusion can all influence rolling and recrystallization texture. For example, it has been observed that grain refinement in the initial microstructure prior to annealing is crucial for the development of the γ-fiber [4,5]. On the other hand, the effect of the initial microstructure on rolling and recrystallization textures was not considered, and should be researched further in the future.

## 4. Conclusions

The influence of cold-rolling directions on the recrystallization texture evolution of pure iron was researched, and the following results were obtained.

(1)The development of γ-fiber and {100}<011> orientation was observed in both cold-rolled specimens. Furthermore, γ-fiber formed comparatively more in specimen A, while α-fiber developed comparatively more in specimen B.(2)The strain distribution in cold-rolled specimen A was presumably inhomogeneous, whereas it was rather uniform at the macro-scale in cold-rolled specimen B.(3)In the case of annealed specimen A, the formation of γ-fiber was confirmed. In annealed specimen B, however, the recrystallization texture tended to be random, and the formation of α-fiber was observed. The randomization of the recrystallization texture was attributed to the uniform strain distribution in cold-rolled specimen B. Furthermore, the formation of α-fiber in cold-rolled specimen B was attributed to continuous recrystallization from severely strained α-fiber.(4)The formation of Goss orientation was established in both specimens. Recrystallized ferrite grains with Goss orientation nucleated in high strain regions of cold-rolled specimen.

## Figures and Tables

**Figure 1 materials-15-03083-f001:**
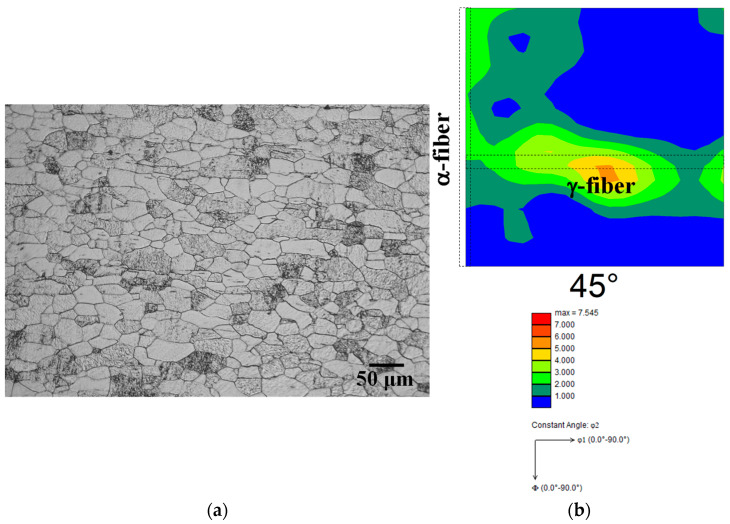
(**a**) Optical micrograph and (**b**) ODF map of as-received sheet.

**Figure 2 materials-15-03083-f002:**
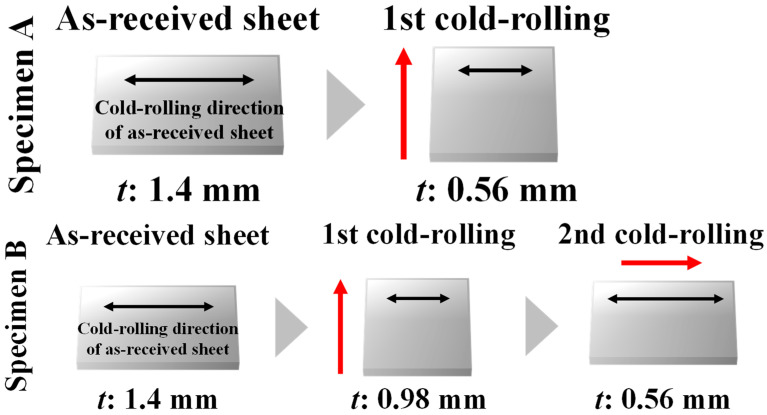
Schematics of cold-rolling conditions (red arrows: cold-rolling direction).

**Figure 3 materials-15-03083-f003:**
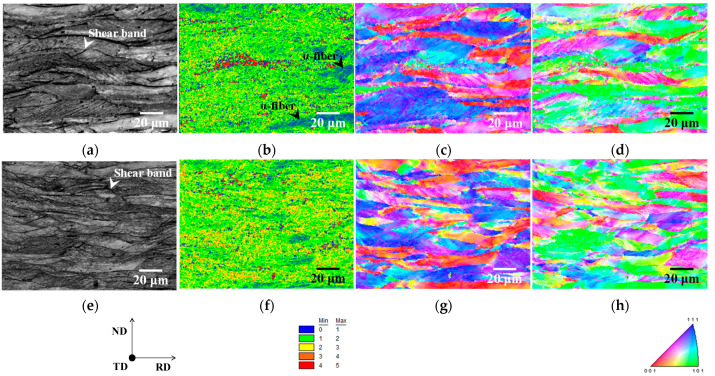
(**a**,**e**) Image quality, (**b**,**f**) kernel average misorientation, and inverse pole figure maps in the (**c**,**g**) normal and (**d**,**h**) rolling directions of cold-rolled specimens A and B.

**Figure 4 materials-15-03083-f004:**
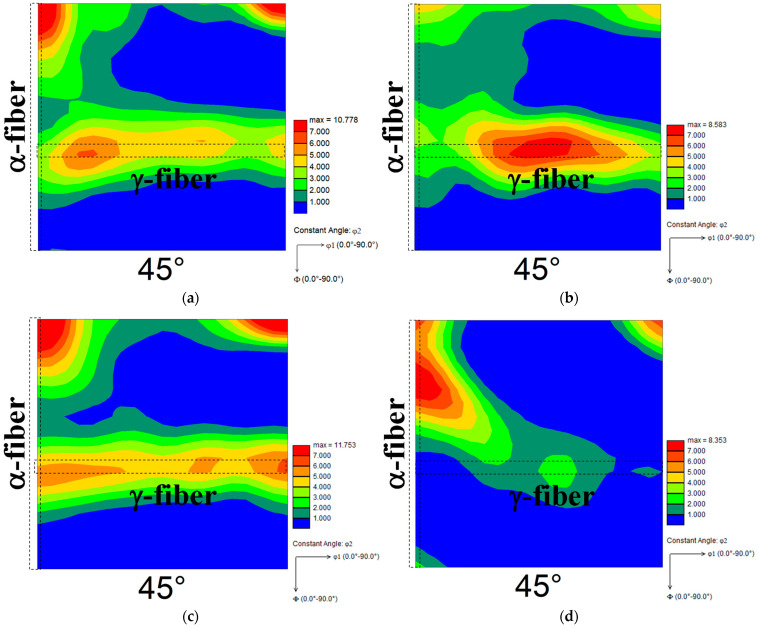
Two typical ODF maps of cold-rolled specimens (**a**,**b**) A and (**c**,**d**) B (φ2 = 45° section).

**Figure 5 materials-15-03083-f005:**
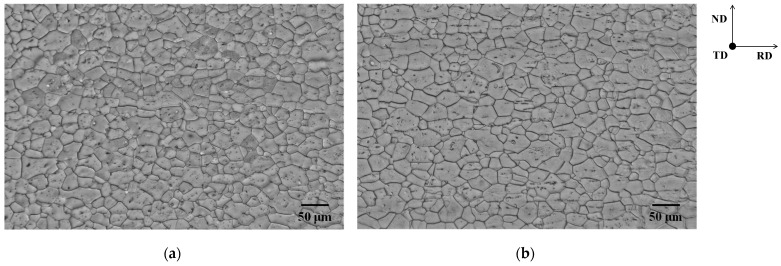
Optical micrographs of specimens (**a**) A and (**b**) B heated to 1073 K.

**Figure 6 materials-15-03083-f006:**
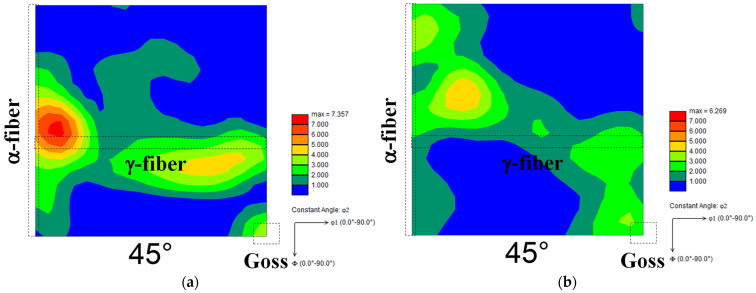
ODF maps of specimens (**a**) A and (**b**) B heated to 1073 K (φ2 = 45° section).

**Figure 7 materials-15-03083-f007:**
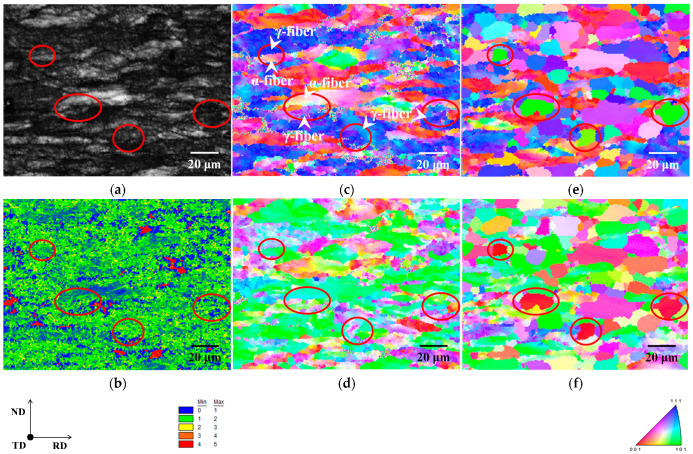
(**a**) Image quality, (**b**) kernel average misorientation, and inverse pole figure maps in the (**c**) normal and (**d**) rolling directions of cold-rolled specimen A and inverse pole figure maps in the (**e**) normal and (**f**) rolling directions of annealed specimen A.

**Figure 8 materials-15-03083-f008:**
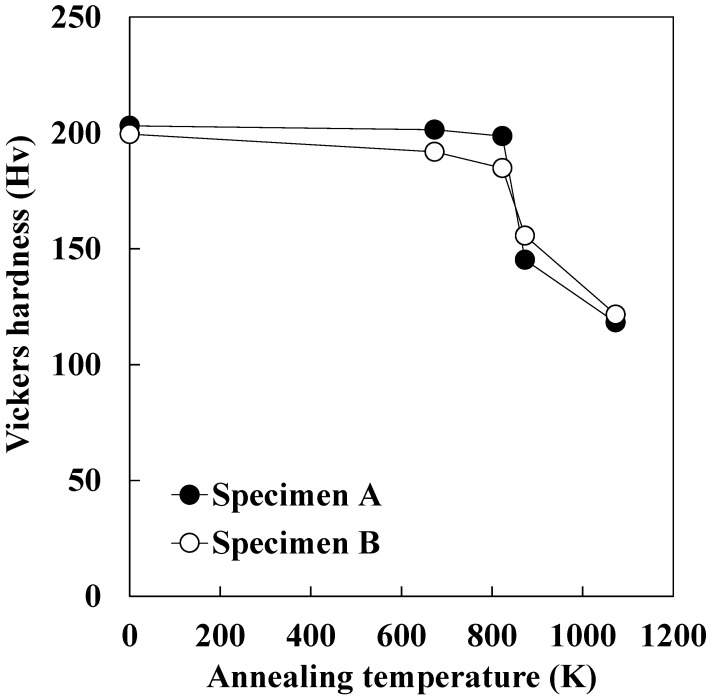
Changes in Vickers hardness during heating to 1073 K.

**Figure 9 materials-15-03083-f009:**
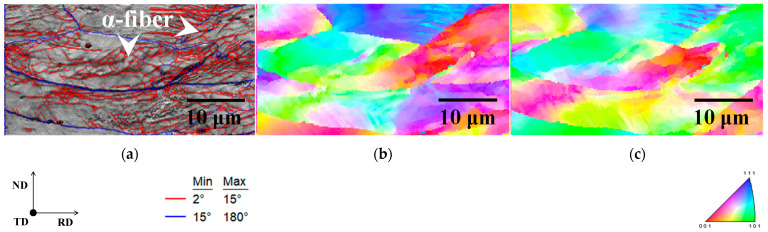
(**a**) Image quality with grain-boundary misorientation angle and inverse pole figure maps in the (**b**) normal and (**c**) rolling directions of specimen B heated to 823 K.

## Data Availability

The data presented in this study are available on request from the corresponding author.

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
