# Peer review of "Effect of Cold-Rolling Directions on Recrystallization Texture Evolution of Pure Iron"

_materials, 2022, doi:10.3390/ma15093083_

Round 1
Reviewer 1 Report
It is in attached file.
Author Response
Response to Reviewer 1 Comments
We thank the reviewers for their careful reading of our paper and their comments. We have amended the paper accordingly. Significant changes have been made (in red in the revised version). Please find a list of modifications and comments below. Additionally, English language and style were checked again by a native speaker.
Comment 1: Line 58 and in the titles of figures 1, 3 and 5 (page 2, 4-5): What is an ODF map? Put at least once the complete words of ODF, and between parentheses (ODF). And then you can use it abbreviated in the rest of the article.
Response: Thank you for your comment. ODF is an abbreviation for orientation distribution function, and we added the explanation.
Comment 2: Lines 71-73 page 2: What is an RD-ND and OIM? Put at least once the complete words of RD-ND and OIM.
Response: Thank you for your comment. RD and ND are abbreviations for rolling direction and normal direction, and this explanation has already been given. OIM is an abbreviation for orientation imaging microscopy, and we added the explanation.
Comment 3: Reference (pages 8-9): References should comply with the regulations of the journal. I enclose the Materials standards.
Response: Thank you for your comment. We corrected the format of references.
Reviewer 2 Report
Ogawa et al present a study of cold-rolled pure Fe sheets, examining the influence of rolling direction and sequence on the development of textures and hardness. The work is interesting and the topic is suitable for materials, however the examination of the cause and effect relationship between rolling and properties is insufficiently developed. In addition, many of the figures need to be clarified to enable easier digestion of the results.
The details of the processing conditions are key to understand the results presented, but they are somewhat poorly explained. I would suggest adding a diagram of rolling directions to fig. 1, as well as indicators of how the rolling direction and the non-rolling direction are defined in the case of specimen B. Further, these conditions should be explained in the abstract to allow the reader searching for information to identify this paper as one of interest. Similarly, it would be helpful to define r-value in the introduction , possibly in the section defining the fibers, to enable the non-specialist reader to follow the work.
When reporting the grain size in the as-received and after rolling and heat treating, it would be useful to have error bars based on the width of the size distribution, or else to directly show the size distribution.
The interpretation provided in lines 98 - 102 is confusing to me. The strain in deformed grains with alpha fiber is minimal in specimen A, but the authors then state that in specimen B the strain is high in deformed grains, and that the specimen is overall more strained. Can the authors explain why this is the case? If all of B is more strained, are the deformed grains in B strained yet more? The authors should link these findings to the rolling conditions to better explain why it has happened.
In Figure 3 two ODF maps are presented for each A and B, but the maps for B (c,d) are rather dissimilar. Does this suggest that the samples are inhomogeneous? How does the presence of very strongly developed gamma fiber in c affect the conclusions? The authors mention only the alpha fiber in the conclusion section, but this is not the only texture observed.
Line 121: "non ferrite grains were rare". Please quantify this rarity, and explain how ferrite vs non-ferrite was assessed.
The discussion of Goss orientation in lines 128 - 135 is lacking. How do the results relate to the details of the cold rolling? Could the process be modified to control its formation? How does this relate to the shear bands?
Figures 2 and 6: how does the orientation of these images relate to the rolling directions?
How have the authors determined the range of recrystallization? And how does this and the hardness data relate to the samples that were held at elevated temperatures for extended periods as discussed in line 68,69? The slow ramp rate to temperature seems surprising given the wide range of temperatures covered, and the very short holds at temperature.
The conclusions should be somewhat expanded to cover more details of why these rolling conditions lead to these results, and how they might be modified to change the findings. Otherwise the paper lacks interpretation and is more of a report.
Minor points
The lables and axes on the ODF figures are too small to read, the details should be enlarged
The optical micrographs need to have rolling directions indicated on them
Line 51 refers to the mechanism of recrystallization texture in an earlier study, but does not mention which study this is. Please add a citation.
Can the authors estimate the cooling rate achieved in water quenching?
Author Response
Response to Reviewer 2 Comments
We thank the reviewers for their careful reading of our paper and their comments. We have amended the paper accordingly. Significant changes have been made (in red in the revised version). Please find a list of modifications and comments below. Additionally, English language and style were checked again by a native speaker.
Comment 1: The details of the processing conditions are key to understand the results presented, but they are somewhat poorly explained. I would suggest adding a diagram of rolling directions to fig. 1, as well as indicators of how the rolling direction and the non-rolling direction are defined in the case of specimen B. Further, these conditions should be explained in the abstract to allow the reader searching for information to identify this paper as one of interest. Similarly, it would be helpful to define r-value in the introduction, possibly in the section defining the fibers, to enable the non-specialist reader to follow the work.
Response: Thank you for your advice. We added a diagram of rolling directions as Fig. 2 and information regarding the cold-rolling conditions in abstract. Additionally, we defined r-value in the introduction.
Comment 2: When reporting the grain size in the as-received and after rolling and heat treating, it would be useful to have error bars based on the width of the size distribution, or else to directly show the size distribution.
Response: Thank you for your advice. We added the standard deviation of grain size.
Comment 3: The interpretation provided in lines 98 - 102 is confusing to me. The strain in deformed grains with alpha fiber is minimal in specimen A, but the authors then state that in specimen B the strain is high in deformed grains, and that the specimen is overall more strained. Can the authors explain why this is the case? If all of B is more strained, are the deformed grains in B strained yet more? The authors should link these findings to the rolling conditions to better explain why it has happened.
Response: Thank you for your advice. We added the average of KAM value in cold-rolled specimens A and B. The average of KAM value in specimen B is larger than that in specimen A, and it means that specimen B is more strained overall than specimen A.
Comment 4: In Figure 3 two ODF maps are presented for each A and B, but the maps for B (c,d) are rather dissimilar. Does this suggest that the samples are inhomogeneous? How does the presence of very strongly developed gamma fiber in c affect the conclusions? The authors mention only the alpha fiber in the conclusion section, but this is not the only texture observed.
Response: Thank you for your advice. As you pointed out, the data shown in Fig. 4(c,d) suggests that the texture is inhomogeneous in cold-rolled specimen B. If γ-fiber strongly develops in cold-rolled specimen B, the randomization of recrystallization texture may not be observed because recrystallization from γ-fiber preferentially occurs.
Comment 5: Line 121: "non ferrite grains were rare". Please quantify this rarity, and explain how ferrite vs non-ferrite was assessed.
Response: Thank you for your advice. We modified the sentence (“no non-recrystallized ferrite grains were observed”).
Comment 6: The discussion of Goss orientation in lines 128 - 135 is lacking. How do the results relate to the details of the cold rolling? Could the process be modified to control its formation? How does this relate to the shear bands?
Response: Thank you for your advice. In this study, cross cold-rolling leads to higher strain, thereby producing shear bands. We added this sentence.
Comment 7: Figures 2 and 6: how does the orientation of these images relate to the rolling directions?
Response: Thank you for your advice. We added the rolling direction in Figs. 3, 7, and 9.
Comment 8: How have the authors determined the range of recrystallization? And how does this and the hardness data relate to the samples that were held at elevated temperatures for extended periods as discussed in line 68,69? The slow ramp rate to temperature seems surprising given the wide range of temperatures covered, and the very short holds at temperature.
Response: Thank you for your advice. We determined the range of recrystallization from both microstructure and hardness. Additionally, the beginning temperature of recrystallization in pure iron has been reported to be approximately 823 K [15], and this is consistent with the result obtained in this study. We added the sentence regarding the beginning temperature of recrystallization.
Comment 9: The conclusions should be somewhat expanded to cover more details of why these rolling conditions lead to these results, and how they might be modified to change the findings. Otherwise, the paper lacks interpretation and is more of a report.
Response: Thank you for your advice. We modified the conclusions.
Comment 10: The labels and axes on the ODF figures are too small to read, the details should be enlarged. The optical micrographs need to have rolling directions indicated on them. Line 51 refers to the mechanism of recrystallization texture in an earlier study, but does not mention which study this is. Please add a citation. Can the authors estimate the cooling rate achieved in water quenching?
Response: Thank you for your advice. We enlarged the labels and axes on the ODF figures. In addition, we added the rolling direction in Fig. 5 and the literature in the introduction. Unfortunately, the cooling rate in water quenching cannot be estimated, but the cooling rate is considered to have little effect on microstructural evolution in this study.
Reviewer 3 Report
The content of the article titled: "Effect of Cold-Rolling Directions on Recrystallization Texture Evolution of Pure Iron" has successfully studied the influence of the directions of the cold rolling method on the crystallization of iron. The results obtained are very interesting which, at present, cannot be published unless the authors make very large edits to the content of the article, through the following recommendations:
- The results obtained at present, have not clarified the nature of the problem to be studied, the authors need to analyze and clarify the following issues:
- It is necessary to elucidate the nature of cold rolling directions to the formation of the crystalline structure of iron crystals, why the author chooses only 2 directions α, γ of the material. So with the other directions, what will happen, how will they affect the crystallization process.
- What causes the change in iron crystal structure at the temperature of 823K, the authors need to clearly explain the nature of this crystal structure change, many results are confirming this. this by other research methods such as simulation or theory.
- The results obtained, it is necessary to compare directly with the results obtained previously to increase the accuracy of the results obtained.
- In the introduction, it is necessary to clarify the process that changes the structure of iron during cold rolling. It is necessary to update and supplement documents from 2016 onwards related to the process of structural change under the influence of factors such as atomic number, heating rate, ... to enrich the content of the article. Write and confirm the accuracy of the cold rolling method.
- The content of the summary, introduction, and conclusion is too sketchy to highlight the meaning of the article.
The authors need to revise the entire layout, focus mainly on the results and discuss by separating it into 2 main parts: Structural features, factors affecting the crystallization process of the material, and then compare and make the optimal conclusion of the α, γ orientation selection method.
- Why did the authors choose to choose the cold rolling method without using other research methods, with this method, some advantages and disadvantages that need to be pointed out.
Why the author chooses to investigate materials at different temperatures, the item aims to clarify the problem, the author needs to clarify this issue. As far as I know, maybe the author is trying to determine the crystallization point of the material. If yes, then the author needs to investigate more points around the temperature range from 800K to 1000K to be able to conclude this accurately or the author can use previously published results to confirm. This choice is determined because with Fe material, the crystallization temperature is about 900 K. To increase the attractiveness of the content of the article, the author can add the following documents related to the influence of factors involved in the crystallization of metals:
- DOI: 10.1039/C6RA27841H
- DOI: 10.1016/j.molstruc.2020.128498
- DOI: 10.3390/ma13163631
- DOI: 10.3390/cryst8120469
- It is recommended that the authors check the entire manuscript and correct all English grammar errors or use a professional English editing service.
Wish the author team success, with this useful work.
Author Response
Response to Reviewer 3 Comments
We thank the reviewers for their careful reading of our paper and their comments. We have amended the paper accordingly. Significant changes have been made (in red in the revised version). Please find a list of modifications and comments below. Additionally, English language and style were checked again by a native speaker.
Comment 1: It is necessary to elucidate the nature of cold rolling directions to the formation of the crystalline structure of iron crystals, why the author chooses only 2 directions α, γ of the material. So with the other directions, what will happen, how will they affect the crystallization process.
Response: Thank you for your advice. Previous research on cross-rolled IF steel has already been described, and unique textures have been obtained by cross-rolling. Based on that, the motivation of this research is that the textures of cross-rolled pure iron are not known, and we attempted to discover the possibility of producing diverse textures by devising cold rolling directions. Additionally, previous research on conventional one-way rolling has already been described, and we modified these sentences.
Comment 2: What causes the change in iron crystal structure at the temperature of 823K, the authors need to clearly explain the nature of this crystal structure change, many results are confirming this. this by other research methods such as simulation or theory.
Response: Thank you for your advice. It is likely that the change in microstructure at the temperature of 823 K is attributed to the recovery progress, and we added the literature regarding the recovery.
Comment 3: It is necessary to compare directly with the results obtained previously to increase the accuracy of the results obtained.
Response: Thank you for your advice. In the case of cold-rolled specimen, the textures obtained in this study is consistent with the finding in the previous study [13]. In the case of annealed specimen, the development of γ-fiber is consistent with the finding in the previous study [4-6]. Additionally, the novelty of this study is the formation of Goss orientation, the mechanism can be explained by the formation of shear bands due to cross-rolling [21-23]. As mentioned above, we are comparing directly with the results obtained previously.
Comment 4: In the introduction, it is necessary to clarify the process that changes the structure of iron during cold rolling. It is necessary to update and supplement documents from 2016 onwards related to the process of structural change under the influence of factors such as atomic number, heating rate, ... to enrich the content of the article. Write and confirm the accuracy of the cold rolling method.
Response: Thank you for your advice. The texture evolution during cross-rolling has already been mentioned. Additionally, we added the literature regarding the texture evolution process under the influence of factors.
Comment 5: The content of the summary, introduction, and conclusion is too sketchy to highlight the meaning of the article.
Response: Thank you for your advice. We modified the abstract, introduction, and conclusions.
Comment 6: The authors need to revise the entire layout, focus mainly on the results and discuss by separating it into 2 main parts: Structural features, factors affecting the crystallization process of the material, and then compare and make the optimal conclusion of the α, γ orientation selection method.
Response: Thank you for your advice. Section 3.1 (Analysis of cold-rolled sheets) was divided into 3.1.1 (Microstructural features of cold-rolled sheets) and 3.1.2 (Features of rolling texture).
Comment 7: Why did the authors choose the cold rolling method without using other research methods, with this method, some advantages and disadvantages that need to be pointed out.
Response: Thank you for your advice. As mentioned above, previous research on cross-rolled IF steel has already been described, and unique textures have been obtained by cross-rolling. Based on that, the motivation of this research is that the textures of cross-rolled pure iron are not known, and we attempted to discover the possibility of producing diverse textures by devising cold rolling directions.
Comment 8: Why the author chooses to investigate materials at different temperatures, the item aims to clarify the problem, the author needs to clarify this issue. As far as I know, maybe the author is trying to determine the crystallization point of the material. If yes, then the author needs to investigate more points around the temperature range from 800 K to 1000 K to be able to conclude this accurately or the author can use previously published results to confirm. This choice is determined because with Fe material, the crystallization temperature is about 900 K. To increase the attractiveness of the content of the article, the author can add the following documents related to the influence of factors involved in the crystallization of metals.
Response: Thank you for your advice. We added a literature to improve the reliability of data.
Comment 9: It is recommended that the authors check the entire manuscript and correct all English grammar errors or use a professional English editing service.
Response: Thank you for your advice. English language and style were checked again by a native speaker.
Round 2
Reviewer 2 Report
The authors have largely addressed my concerns, the manuscript is suitable for publication. A final polish of the conclusion to clarify underlying mechanisms and future study directions would be helpful but is not essential.
Reviewer 3 Report
The authors need to answer in detail, clarify the previous questions in manuscript content, need to add more analyses and point out the nature of this cold rolling method to highlight the content of the article.
After additional editing, checking the entire English grammar style to meet the requirements of the journal, reviewers accept publishing and authors do not need to send reply back.
Congratulations on the success of the author team with this useful work.